# MULTI-STEP PREDICTIVE LEARNING LEADS TO SIMPLICITY BIAS

## ABSTRACT

Predictive learning is a framework for understanding the formation of low-dimensional internal representations mirroring the environment's latent structure. The conditions under which such representations emerge remain unclear. In this work, we investigate how the prediction horizon and network depth shape the solutions of predictive learning tasks. Using a minimal abstract setting inspired by prior work, we show empirically and theoretically that sufficiently deep networks trained with multi-step prediction horizons consistently recover the underlying latent structure, a phenomenon explained through the Ordinary Least Squares estimator structure and biases in learning dynamics. We then extend these insights to nonlinear networks and complex datasets, including piecewise linear functions, MNIST, multiple latent states and higher dimensional state geometries. Our results provide a principled understanding of when and why predictive learning induces structured representations, bridging the gap between empirical observations and theoretical foundations.

## 1 INTRODUCTION

Predictive coding has emerged as a powerful theoretical framework for understanding both learning and representation in neural networks. At its core, predictive coding posits that neural networks continuously construct and refine internal models of their inputs to minimize prediction error of incoming stimuli. This idea has gained significant traction as a unifying principle for perception, action, and learning (Friston, 2010). In machine learning, predictive coding have been used as a form of unsupervised learning (Wen et al., 2018; Lotter et al., 2016). In neuroscience, predictive learning tasks have been used to study how networks build *world models*—internal representations that capture the latent structure of the environment—by requiring them to predict future inputs given the past (Recanatesi et al., 2021). Predictive coding and predictive learning are conceptually closely related. The former constitutes a biologically plausible learning rule, while the latter constitutes a biologically plausible unsupervised task demand.

Despite these advances, several fundamental questions remain unresolved. While many studies report that predictive learning networks develop low-dimensional, interpretable representations of latent variables, this outcome is not guaranteed. Intuitively, learning a latent world model seems advantageous for predictive tasks, as structured representations may enable more efficient predictions. However, overparameterized networks admit infinitely many solutions that can achieve perfect performance without forming any interpretable representation (Frankle & Carbin, 2018; Zhang et al., 2016; Nguyen & Hein, 2017). The mere success of predictive learning at minimizing prediction error therefore does not explain why structured world models emerge in practice. A deeper understanding of the inductive biases introduced by prediction horizon, network depth, and training dynamics is needed to clarify *when* and *why* predictive learning induces meaningful internal representations.

Recent work shows that the *prediction horizon*—how far into the future the network must predict—plays a critical role in shaping the learned representations (Levenstein et al., 2024; Vollan et al., 2025). Here, we take a first step toward understanding this effect and the general rules of predictive representations by constructing a minimal linear predictive learning problem inspired by Recanatesi et al. (2021). This setting is analytically tractable yet rich enough to capture essential aspects of the problem. We show empirically that when the network is sufficiently deep and the prediction horizon scales linearly with the environment size, gradient descent consistently con-

verges to highly structured solutions that recover the underlying state from the observations. To explain this phenomenon, we combine tools from machine learning theory with an analysis of the Ordinary Least Squares estimator structure, revealing how the prediction horizon and loss choice (cross-entropy rather than mean squared error) shape the solution landscape.

Building on this intuition, we then extend our study to more complex settings, including nonlinear networks, continuous environments and stochastic observations, as well as settings with multiple independent environments. Across these experiments, we test the generality of the principles uncovered in the linear case, exploring how task structure, training biases, and prediction horizon interact to produce ordered representations.

Our contributions are threefold:

1. **Empirical characterization** of when predictive learning induces state representations in linear networks.

2. **Theoretical analysis** linking OLS estimator structure and training dynamics biases to representation learning.

3. **Extension to nonlinear and more natural settings**, demonstrating the robustness of the observed phenomena to modeling choices.

Together, these results advance our understanding of how predictive learning interacts with model architecture, task design, and optimization dynamics to shape internal representations.

## 2 RELATED WORK

Predictive learning has been shown to uncover latent structure in environments. Recanatesi et al. (2021) demonstrated that predictive learning can recover low-dimensional latent spaces in discrete, continuous, and angular settings. However, they did not examine when such structure fails to emerge. Levenstein et al. (2024) extended this line of work, showing that recurrent networks form continuous attractors under multi-step but not next-step prediction, underscoring the role of prediction horizon. Our study builds on these findings by analyzing how horizon length, network depth, and optimization dynamics bias predictive learning solutions.

Previous works have proposed several approaches for extracting latent structure within predictive frameworks. Watter et al. (2015) introduced a model that enforces locally linear latent dynamics through an explicit architectural prior. Saanum et al. (2024) encouraged simplified latent dynamics by imposing a soft state-invariance regularizer, biasing the latent state to change slowly unless driven by actions. Kipf et al. (2019) learned structured latent transitions using a contrastive objective that separates true next states from negatives. While these methods demonstrate that predictive learning can reveal aspects of latent geometry, they rely on architectural constraints, explicit regularization, or assumed structure of the environment. In contrast, our work provides a mechanistic explanation for why and when multi-step predictive learning alone—without additional regularization—reshapes the data geometry and consistently drives networks toward representations that recover the underlying latent state.

Separately, theoretical results on implicit bias in classification show that gradient descent converges to the hard-margin SVM solution for linearly separable data. This has been established for single-layer (Soudry et al., 2018), deep linear (Ji & Telgarsky, 2018), and homogeneous networks (Lyu & Li, 2019). We use these results to characterize the implicit bias of deep linear networks in our abstract predictive learning classification task, and make a connection that was previously overlooked: In deep neural networks performing multiclass classifications, the parameters converge to the hard margin SVM with regularization over the weight matrix rank rather than its $L_2$ norm.

Finally, while related to Neural Collapse (Papyan et al., 2020), our findings differ: in our work representations collapse toward the latent geometry of the environment rather than a simplex, indicating that the effect arises from the structure of the environment rather than an optimal decoding geometry.

## 3 RESULTS

We begin by formalizing the task of multi-step predictive learning in a simple setting (Figure 1).

$$\boldsymbol{x} = (O(s), g(a)), \quad \boldsymbol{y} = O(s + a), \quad s \in [1, S], \ a \in [-A, A].$$

Here, $O$ and $g$ denote high-dimensional observation functions parameterized by lower-dimensional states $s$ and actions $a$. $S$ denotes the number of states. The parameter $A$ specifies the maximal action range, which can be interpreted as the maximal or typical trajectory length considered by the model. This formulation abstracts away explicit time dependence: instead of modeling continuous temporal evolution, it is equivalent to sampling from a memory buffer of state–action trajectories. For completeness sake, we include in the appendix simulations of Recurrent Neural Networks performing k-step prediction in a gridworld, similar to Levenstein et al. (2024). We will first consider a deterministic and discrete environment in which each state maps to a single observation. These assumptions will be relaxed when extending to more complex and naturalistic settings.

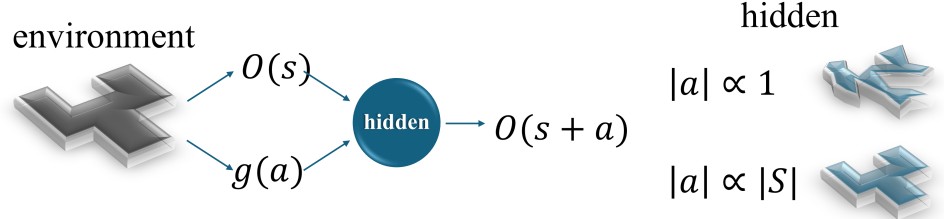

Figure 1: Illustration of multi-step predictive learning setting where time is abstracted away. An agent is acting in an environment, producing a set of observations and actions in its trajectory. The task is to predict, for each action and observation pair, the following observation. The environment has an underlying structure, and training a model on a predictive learning task sometimes generates a representation of this latent structure. Recent work has shown that increasing the prediction horizon can lead to more accurate and stable representations (Levenstein et al., 2024).

### 3.1 SPONTANEOUS COLLAPSE TO ORDER IN MULTI-STEP ABSTRACT PREDICTIVE LEARNING

We consider an abstract predictive learning task inspired by Recanatesi et al. (2021), in which $S$ states and $2A + 1$ actions are represented by one-hot encoded observations $O(s) = \delta_s \in \mathbb{R}^S$, $s \in [1, S]$, $g(a) = \delta_a \in \mathbb{R}^{2A+1}$, $a \in [-A, A]$. $A$ specifies the maximal allowed action. The network receives inputs of the form $\{O(s), g(a)\}$. The target output is the next observation $O(s+a)$. Tuples that map to undefined states are discarded. We train a deep linear network with $L$ layers and no biases on the full dataset of all possible state–action pairs. Since the observations are one-hot encoded, training with cross-entropy loss naturally casts the task as multiclass classification.

Our goal is to study the relationship between the environment's latent geometry and the network's internal representation. The former is given by the shifted state $s+a$, while the latter we define as the activation of the last hidden layer. Crucially, the choice of one-hot encoding removes any intrinsic correlations between neighboring states. Thus, any emergent structure in the hidden representations must arise from learning to solve the predictive task.

In an overparameterized network, there exist infinitely many perfect accuracy solutions for any given $A$. For example, a solution for a large $A$ trivially satisfies the task for smaller $A$. The key question is therefore: which solution does the network converge to, and why? Figure 2 shows that in the multi-step setting, hidden activations spontaneously collapse onto the latent state manifold, whereas in the single-step setting they do not. This emergence of consistent and highly ordered representations in an unconstrained optimization problem highlights the presence of strong implicit biases in the training dynamics. Although disorganized solutions are equally valid, the network reliably converges to the collapsed manifold structure—but only when the prediction horizon scales linearly with $S$ (Figure S2). As we will show, explaining this phenomenon requires combining several theoretical results with a deeper analysis of the problem structure.

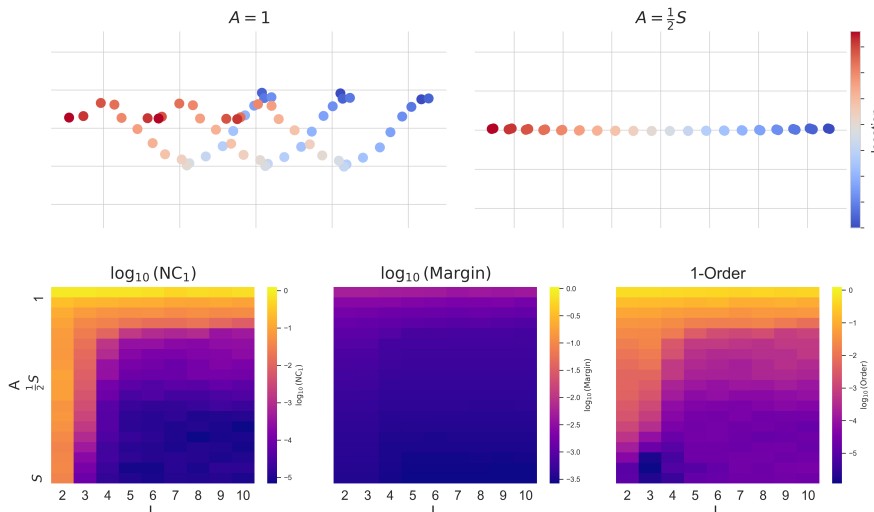

Figure 2: Top: the first two principal components of hidden activations for networks trained on single-step prediction (left) and multi-step prediction with maximal action $A = S/2$ (right). Bottom: quantitative metrics across values of $A$ and network depth $L$. Left—NC1 decreases with increasing $A$ and $L$, indicating more compact class clusters. Middle—normalized margins (relative to $L = 2$) decrease with depth. Right—representations become increasingly aligned with the target state as $A$ and $L$ grow. See the appendix for detailed metric definitions. The analysis was done for $|a| <= 1$.

## 3.2 MECHANISM OF COLLAPSE TO ORDER

Because the task is classification with a linear network, prior work shows that training should converge to the maximal margin solution (Soudry et al., 2018; Ji & Telgarsky, 2018; Lyu & Li, 2019). However, as illustrated in Figure 2, networks of different depths trained on the same task achieve different functional margins. At first glance, this appears to contradict the theoretical results. To resolve this discrepancy, we turn to the concept of *representation cost* (Dai et al., 2021).

Specifically, Lyu & Li (2019) showed that a deep linear network trained on multiclass data converges to the parameters that solve

$$\underset{\boldsymbol{W}_1,\ldots,\boldsymbol{W}_L}{\arg\min} \sum_{l=1}^{L} \|\boldsymbol{W}_l\|_2^2 \quad \text{s.t.} \quad \forall i, \forall k \neq y_i : \boldsymbol{W}_{y_i}^\top \boldsymbol{x}_i \geq \boldsymbol{W}_k^\top \boldsymbol{x}_i + 1,$$

where $\boldsymbol{W} = \prod_{l=1}^{L} \boldsymbol{W}_l$. Although this resembles the hard-margin multiclass SVM, it is not identical. Dai et al. (2021) further showed that in deep linear networks, $L_2$ regularization on the parameters corresponds in functional space to a Schatten $2/L$ quasi-norm. Thus, in functional space the optimization problem becomes

$$\underset{\boldsymbol{W}}{\arg\min} \|\boldsymbol{W}\|_{2/L}^{SC} \quad \text{s.t.} \quad \forall i, \forall k \neq y_i : \boldsymbol{W}_{y_i}^\top \boldsymbol{x}_i \geq \boldsymbol{W}_k^\top \boldsymbol{x}_i + 1.$$

This characterization implies that increasing network depth induces a trade-off between effective rank and functional margin: deeper networks bias the solution toward lower-rank approximations of the hard-margin solution at the cost of smaller margins. This explains the empirical observation that both rank and functional margins decrease with depth (Figure 3).

At this point we understand that deep networks are biased towards a low rank approximation of the maximal margin solution, but that still doesn't explain the difference between prediction horizons.

We hypothesize that increasing the prediction horizon adds more constraints to the solution space, narrowing it to a sub-space where states are correlated. We thus investigate the structure of the OLS estimator $\Sigma = (\boldsymbol{X}^T\boldsymbol{X})^{-1}\boldsymbol{X}^T\boldsymbol{Y}$. Intuitively, this matrix can be related to classification when the data is highly balanced and symmetric. In such a case, the OLS estimator can also separate the data into classes. As shown in Figure 3, for larger values of $A$ the OLS matrix becomes effectively lower-dimensional and exhibits two dominant singular values. The leading singular vector corresponds to a linear combination of the input state and action. Importantly, as the prediction horizon increases, this direction explains a growing fraction of the variance in the data, making it increasingly useful for classification. Consequently, networks consistently converge to solutions that exploit this direction.

To build intuition, we analyze the structure of the OLS estimator. The matrix $\boldsymbol{X}^\top\boldsymbol{X}$ has a *block structure* consisting of two diagonal blocks and two off-diagonal *band matrices*. The width of these bands grows linearly with the prediction horizon $A$: for a one-dimensional environment with $S$ states, the non-zero entries in the off-diagonal blocks are confined to a band of width $2A+1$. For small $A$, these bands are very narrow and nearly diagonal, while for large $A$ they become wide and strongly overlapping, effectively coupling many distant states.

Similarly, the cross-term $\boldsymbol{X}^\top\boldsymbol{Y}$ also consists of two banded blocks, with the same width scaling. Thus, as $A$ increases, both $\boldsymbol{X}^\top\boldsymbol{X}$ and $\boldsymbol{X}^\top\boldsymbol{Y}$ become increasingly dense, introducing strong correlations between distant states and actions. This growing overlap causes the spectrum of the OLS estimator to compress, leaving only a few dominant singular directions.

Because directions that capture the majority of the data variance are also the most effective for separating classes, the dominant singular vector of $\Sigma$ naturally becomes the most informative feature for solving the predictive task. This explains why, in the multi-step setting, deep networks consistently converge toward this leading direction, yielding ordered, low-dimensional representations of the environment. In contrast, in the single-step setting, the variance is more spread between directions and thus no single direction separates the data well.

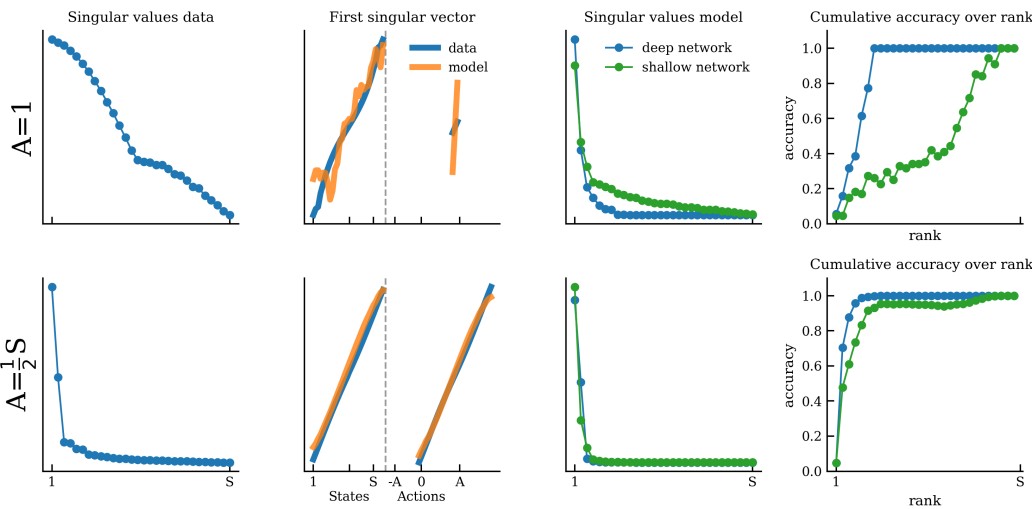

Figure 3: Analyzing the singular values and vectors from the OLS estimator and the model's effective weight matrix. As can be seen, for larger $A$ both become lower dimensional, and the leading singular vector becomes the transformation from the input state and action to the output state. For $A = 1$, since there is no strong direction that explains most variance, the model's singular vectors are mostly decoupled from those of the OLS estimator. We also a comparison between a shallow network ($L = 2$) and a deep network ($L = 9$). As can be seen in the rightmost column, shallow networks depends on directions of the input space to classify the data.

Finally, we can put all the puzzle pieces together and ask whether we gained any intuition for a more general setting. For example, consider a predictive task with two distinct environments, each equipped with its own encoding of actions and states. In this case, the singular vectors of the OLS estimator associated with the two environments are orthogonal. However, because deep

networks exhibit a low-rank bias, we expect training to favor solutions that align the two representations in a way that minimizes the number of active singular values. Figure 4 illustrates this phenomenon. In both tasks, the network embeds the environments within a shared representation space, but under the multi-step objective the resulting structure captures the underlying geometry and introduces a symmetry between analogous representation objects. These results generalize beyond the one-dimensional case, holding also for higher-dimensional latent states, as illustrated in the two-dimensional setting of Figure S1.

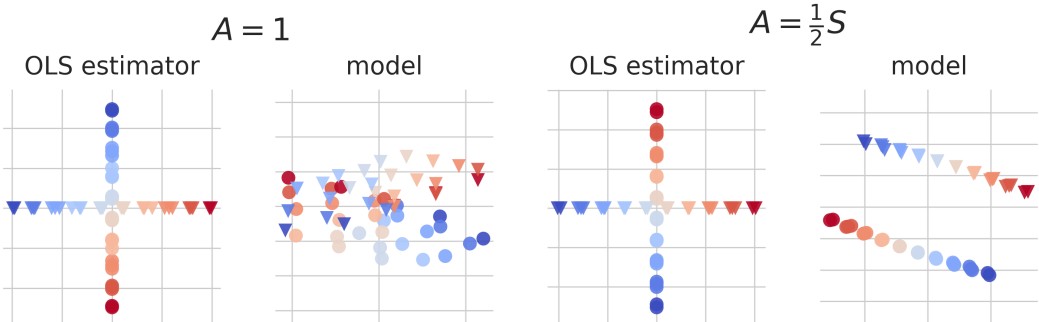

Figure 4: Results for the task with two independent environments. Data are projected onto the first two singular vectors of the OLS estimator, confirming that observations and actions from each environment are orthogonal. In the single-step setting, the model representations remain unaligned, whereas in the multi-step setting they collapse into a shared low-rank structure that aligns the two environments and reveals their underlying symmetry.

### 3.3 GENERALIZATION ACROSS SETTINGS

Next, we consider a more naturalistic setting where the environment is continuous and observations are temporally correlated. Consider a scenario in which an agent moves between several rooms. Inside every room, the environment changes smoothly, while a transition between rooms introduces discontinuities. A reliable representation of such an environment should use prediction to stitch the various rooms together into a coherent world map. In this setup, states are drawn from a uniform distribution $s \sim \mathcal{U}(-1, 1)$, and actions are drawn from a Gaussian distribution $a \sim \mathcal{N}(0, A)$. The observations $O(s)$ capture the world structure. As shown in Figure 5, for small $A$ the network merely mirrors the local autocorrelation of the data, whereas for larger horizons it "stitches" together the different linear segments into a coherent one-dimensional manifold.

To test whether our observations extend to settings where observations are stochastic rather than deterministic, we designed a novel variant of the MNIST task. In this setting, the input to the network is an MNIST digit along with a one-hot encoded action vector, and the target output is the MNIST digit whose label corresponds to the sum of the input digit label and the action. As in the abstract task, we control the maximal allowed action. We trained a Generative Adversarial Network on this task and analyzed the encoder's latent space. The network successfully generates the correct digits in both the single-step and multi-step settings (see appendix). Strikingly, in the multi-step case the latent space organizes along a one-dimensional manifold, where different positions correspond to digits ordered by their labels (Figure 6). By sampling along the first principal component of the latent space, we can generate digits in sequential order, demonstrating that multi-step predictive learning induces structured and interpretable representations even in this more naturalistic dataset. Furthermore, we demonstrate that training with a range of standard regularization methods fails to recover this structure in the single-step setting (Figure S3). Note that we only trained with a few hundred samples per class. When within-class variability is too high, the network does not reliably recover the latent structure. Exploring the reasons for this, as well as identifying bounds on the tolerated variability, is an interesting direction for future work. The abstract framework developed here provides a principled way to study such effects, for example by systematically controlling the noise level in the observations.

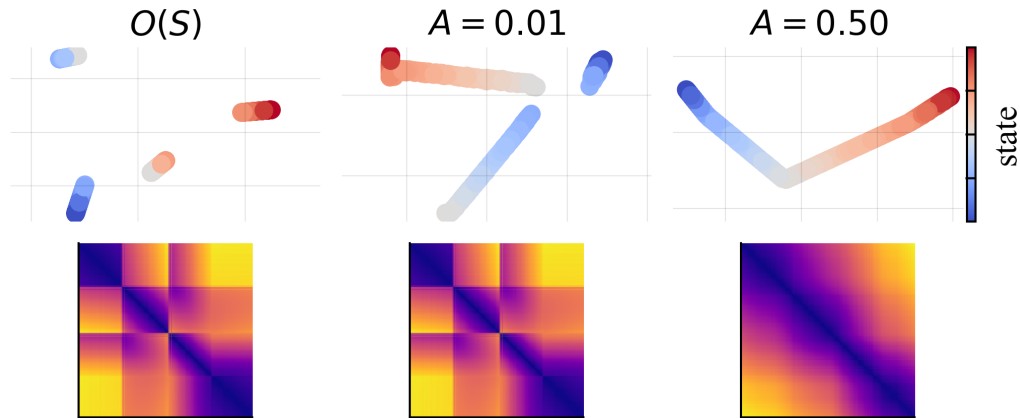

Figure 5: Training a deep nonlinear network on a predictive task with observations generated from a piecewise linear function containing three discontinuities. When the action distribution is narrow, the learned representations primarily mirror local autocorrelation. In contrast, with a wider action distribution, the network organizes its hidden representations along a smooth one-dimensional manifold that bridges the discontinuities, thereby recovering the underlying latent state. The top figure shows the Principle Component Analysis (PCA) space, and the bottom figure shows the distance matrix sorted by the state variable.

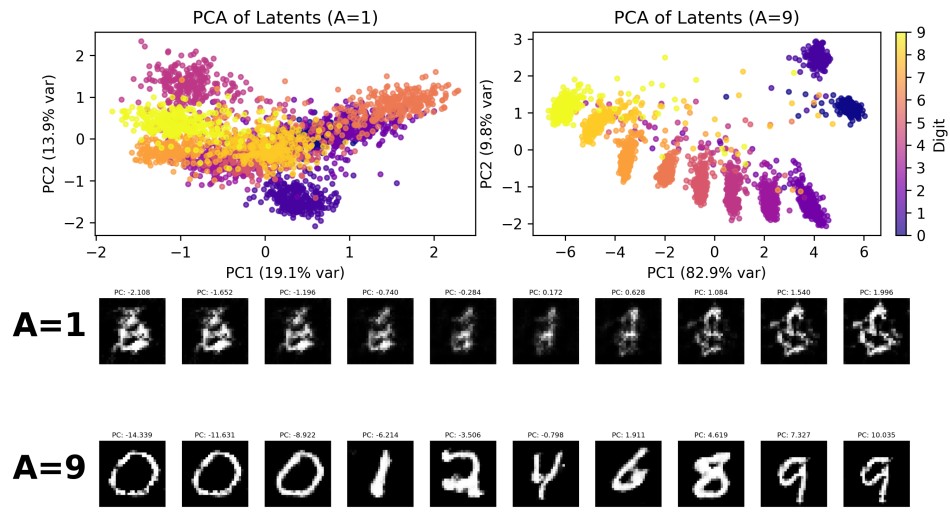

Figure 6: Top: latent activations of the encoder module. In the multi-step setting with $A = 9$, the latents collapse onto a low-dimensional manifold ordered by digit label, whereas in the single-step case no such structure emerges. Bottom: digits generated by sampling along the first principal component of the latent space. In both cases, generation quality is high and the digits are correctly produced (see Appendix).

We now consider a natural setting for this predictive learning task: Predictive Coding Networks (PCNs), which offer a biologically plausible mechanism for such learning. We trained a PCN with 7 layers in the simplified, abstract discrete environment while varying the prediction horizon, observing the same qualitative results as in our previous experiments (Figure 7). Namely, for short

prediction horizons and shallow network depths, representations remain unstructured. Conversely, in deep networks with long prediction horizons, the representations emerge to mirror the underlying line geometry of the environment. A detailed description of these simulations is provided in the appendix.

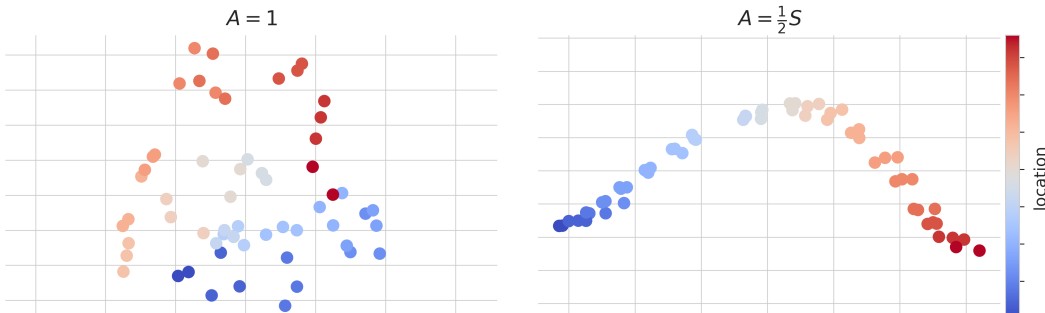

Figure 7: Predictive Coding Network trained on the discrete state abstract prediction task. For single-step predictions, representations have no clear structure while for multi-step prediction, representations are arranged along a line in the two first Principle Components.

## 4 DISCUSSION

We have shown that multi-step predictive learning, together with network depth, acts as a strong inductive bias that drives networks toward low-dimensional, structured representations of the environment's latent variables. Increasing the prediction horizon makes the task more constrained, revealing a dominant direction in the data that deep networks, biased toward low-rank solutions, naturally align with. This explains why structured solutions consistently emerge in the multi-step setting, even when many trivial solutions are possible.

However, several open questions remain. It is unclear why this dominant direction takes such a highly structured form and how this intuition extends to more complex, nonlinear settings. Our experiments on continuous environments, gridworld RNNs and MNIST suggest that similar principles apply, but a full theoretical understanding will require future work. These findings also raise broader implications for machine learning and neuroscience, suggesting that longer prediction horizons may play a key role in the emergence of interpretable world models.

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

## A APPENDIX

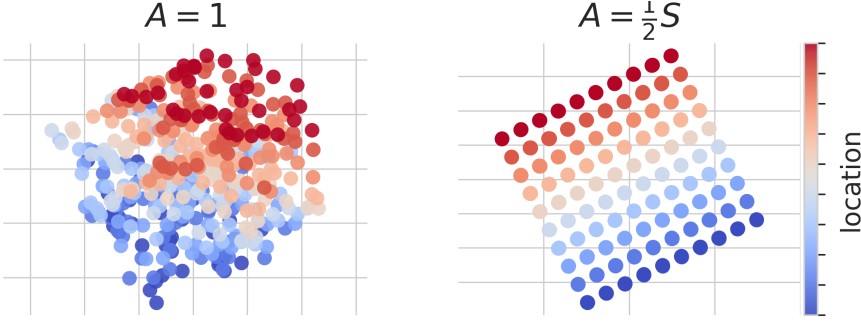

Figure S1: Same abstract task as the linear case, but for a two-dimensional state variable.

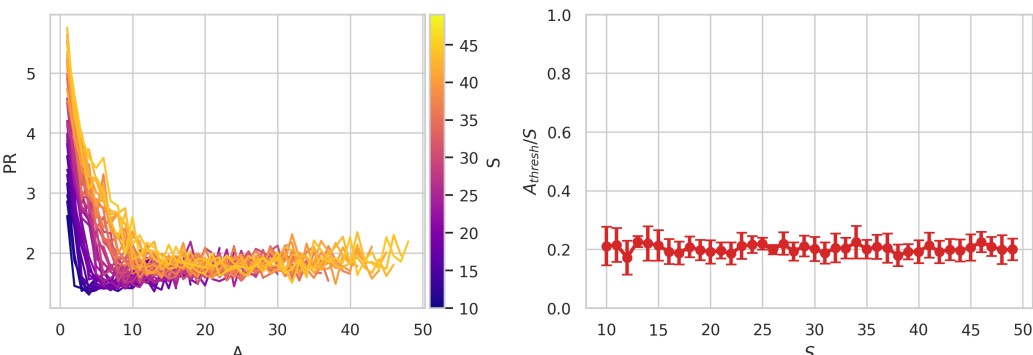

Figure S2: We swept across values of $S$ and $A$ and trained deep nonlinear networks on the predictive task with mean squared error loss. For each $(S, A)$ pair we trained 10 networks and plotted the median participation ratio (PR) of the hidden activations. We define $A_{\text{thresh}}$ as the smallest horizon for which PR drops below 2. As shown, $A_{\text{thresh}}$ scales linearly with $S$, supporting the claim that the prediction horizon required for latent state extraction grows proportionally with the environment size. Error bars are standard deviation obtained by bootstrapping.

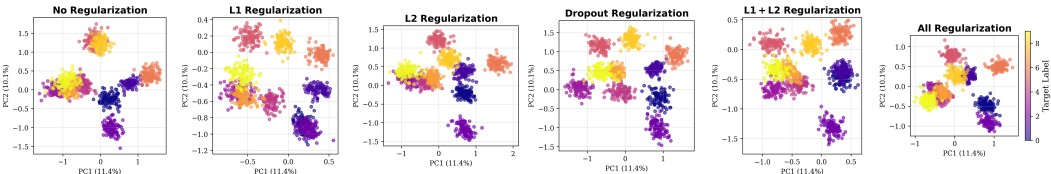

Figure S3: Comparison of multiple regularization types for the single-step case in the MNIST task. Note that no type of regularization produces representations that recover the latent structure as multi-step prediction.

## A.1 CODE AVAILABILITY

Code for running the simulations and generating the figures is attached to this submission. A publicly accessible GitHub repository will be made available in the future.

## A.2 LLM USAGE

LLMs were used to polish the text in the paper, generate code for running simulations, as well as mathematical and technical descriptions in the appendix.

## A.3 METRICS

### A.3.1 NC1

This metric was introduced in Papyan et al. (2020). Given hidden activations

$$h \in \mathbb{R}^{M \times N}, \quad y \in \{1, \ldots, C\}^n,$$

where $M$ is the number of samples, $N$ is the hidden dimension, and $C$ is the number of classes, we define:

$$\mu = \frac{1}{M} \sum_{i=1}^{M} h_i$$

as the global mean of activations, and

$$\mu_c = \frac{1}{m_c} \sum_{i:y_i=c} h_i, \quad m_c = |\{i : y_i = c\}|$$

as the class-conditional means.

The within-class scatter matrix is

$$S_W = \frac{1}{M} \sum_{c=1}^{C} \sum_{i:y_i=c} (h_i - \mu_c)(h_i - \mu_c)^\top,$$

and the between-class scatter matrix is

$$S_B = \frac{1}{M} \sum_{c=1}^{C} m_c (\mu_c - \mu)(\mu_c - \mu)^\top.$$

The NC1 metric is then defined as

$$\text{NC1} = \frac{\text{Tr}(S_W)}{\text{Tr}(S_B)}.$$

Intuitively, $S_W$ captures the variance of samples around their respective class means, while $S_B$ captures the variance of class means around the global mean. The ratio NC1 therefore measures the relative tightness of clusters to their separation: smaller values indicate more compact class representations.

### A.3.2 MARGINS

Let $f : \mathbb{R}^d \to \mathbb{R}^C$ denote the network output function, where

$$f(x) = (f_1(x), f_2(x), \ldots, f_C(x))$$

are the class scores (logits) for input $x \in \mathbb{R}^d$. For each sample $(x_i, y_i)$ with true label $y_i \in \{1, \ldots, C\}$, the functional margin is defined as

$$\gamma_i = f_{y_i}(x_i) - \max_{j \neq y_i} f_j(x_i).$$

The multiclass margin for the dataset is then given by

$$\gamma = \min_{i=1,\ldots,n} \gamma_i.$$

Intuitively, $\gamma_i$ measures the difference between the score assigned to the correct class and the highest score among all incorrect classes for sample $i$. The overall margin $\gamma$ is the worst-case (smallest) of these values across all samples, and therefore characterizes the minimal separation achieved by the classifier.

### A.3.3 $\text{PC}_1^{\text{ORDER}}$

Let $\boldsymbol{H} \in \mathbb{R}^{M \times N}$ denote the hidden activations, where $M$ is the number of samples and $N$ the hidden dimension. We compute the first principal component $\boldsymbol{u}_1 \in \mathbb{R}^N$ of $\boldsymbol{H}$, i.e. the unit-norm eigenvector of the sample covariance matrix

$$\Sigma = \frac{1}{M} \sum_{i=1}^{M} (\boldsymbol{h}_i - \bar{\boldsymbol{h}})(\boldsymbol{h}_i - \bar{\boldsymbol{h}})^\top$$

corresponding to the largest eigenvalue, where $\bar{\boldsymbol{h}} = \frac{1}{M} \sum_{i=1}^{M} \boldsymbol{h}_i$.

Each sample is then projected onto this direction:

$$z_i = \boldsymbol{u}_1^\top (\boldsymbol{h}_i - \bar{\boldsymbol{h}}), \quad i = 1, \ldots, M.$$

Let $s_i \in \mathbb{R}$ denote the state variable associated with sample $i$. The *first PC order* metric is defined as the coefficient of determination ($R^2$) of the linear regression between $\{z_i\}$ and $\{s_i\}$:

$$\mathrm{PC}_1^{\mathrm{order}} = R^2(z, s).$$

This metric measures how strongly the first principal component of the hidden representations aligns with the state. High values of $\mathrm{PC}_1^{\mathrm{order}}$ indicate that the dominant axis of variation in the representation space reflects the state. For D-dimensional states variables we simply take the first D principal components.

### A.3.4 ALIGNMENT

To quantify whether the network aligns representations of separate state variables, we compute an alignment score between subspaces spanned by the leading principal components of their activations. This method is adapted from Sorscher et al. (2022).

Let $\boldsymbol{H}^{(1)},{}^{(2)} \in \mathbb{R}^{M \times N}$ denote hidden activations corresponding to two distinct dataset partitions (e.g., two environments or contexts). For each partition, we compute the sample covariance

$$\Sigma^{(k)} = \frac{1}{M} \sum_{i=1}^{M} \big(\boldsymbol{h}_i^{(k)} - \bar{\boldsymbol{h}}^{(k)}\big)\big(\boldsymbol{h}_i^{(k)} - \bar{\boldsymbol{h}}^{(k)}\big)^{\top}, \quad k \in \{1, 2\},$$

where $\bar{\boldsymbol{h}}^{(k)}$ is the mean activation of partition $k$.

From $\Sigma^{(k)}$, we extract the top principal directions $\boldsymbol{U}^{(k)} \in \mathbb{R}^{N \times m}$, where $m$ is the smallest number of eigenvectors explaining at least a fixed proportion of variance (e.g., 95%).

The alignment score is based on the principal angles $\theta_1 \leq \cdots \leq \theta_m$ between the two subspaces spanned by $\boldsymbol{U}^{(1)}$ and $\boldsymbol{U}^{(2)}$. We take the cosine of the smallest principal angle:

$$\mathrm{Align}(\boldsymbol{H}^{(1)}, \boldsymbol{H}^{(2)}) = \cos(\theta_1).$$

This score lies in $[0, 1]$, with higher values indicating stronger alignment (i.e., the leading directions of variability in the two partitions are closely matched).

## A.4 MNIST EXPERIMENT DETAILS

### A.4.1 MODEL ARCHITECTURE

We implement a conditional Generative Adversarial Network (GAN) for MNIST digit generation with action-based transformations. The model consists of three components:

1. **Encoder** $E(x, a)$: maps input image $x$ and action vector $a$ to a latent representation $z$.

2. **Generator** $G(z)$: maps the latent vector $z$ to an output image.

3. **Discriminator** $D(x)$: classifies images into 11 classes (digits 0–9 plus a "fake" class 10).

### A.4.2 DATASET

We use the MNIST dataset with balanced sampling of $N = 200$ examples per digit class. Actions are integers in the range $[-A, A]$ with $A = 5$, encoded as one-hot vectors of dimension $2A+1 = 11$. Target labels are computed as

$$\mathrm{target} = \mathrm{input\_label} + a.$$

### A.4.3 TRAINING

The model is trained using the Adam optimizer with learning rate $2 \times 10^{-4}$, batch size 64, and 20 epochs. The loss function combines:

1. **Adversarial loss:** the discriminator classifies real images by their true labels and fake images as class 10.

2. **Generator loss:** encourages generated images to be classified as their target labels rather than as fake.

3. **Optional losses:** reconstruction loss (weight $\lambda = 0$ in our experiments) and feature matching loss.

We an adaptive discriminator training ratio, and optional learning rate scheduling for training stability.

### A.4.4 ARCHITECTURE DETAILS

- **Encoder:** Convolutional layers ($28 \times 28 \rightarrow 14 \times 14 \rightarrow 7 \times 7 \rightarrow 3 \times 3 \rightarrow 1 \times 1$) followed by action encoding via an MLP, then combined in a 2-layer MLP with hidden dimension 512.

- **Generator:** 2-layer MLP (latent_dim $\rightarrow 512 \rightarrow 1024 \rightarrow 7 \times 7 \times 128$) followed by transposed convolutions ($7 \times 7 \rightarrow 14 \times 14 \rightarrow 28 \times 28$).

- **Discriminator:** CNN backbone with 11-class classification head.

All networks use LeakyReLU activations and batch normalization.

### A.5 RNN WITH GRIDWORLD

We designed a long, narrow gridworld environment to study sequential prediction in navigation tasks with repeating visual patterns. The environment consists of a $10 \times 2$ grid with a distinct color band that changes along the horizontal axis, creating a corridor-like structure.

The agent receives egocentric observations through a $5 \times 5$ window centered on its current position. The observation space includes:

- One-hot encoded color channels for the colors

- A wall channel indicating out-of-bounds areas

- An object channel for randomly placed objects outside the grid boundaries

The total observation dimension is $d_{\text{obs}} = 5 \times 5 \times (10 + 2) = 300$. Additionally, 10 randomly placed objects are positioned outside the grid boundaries within a margin of 2 cells to provide additional visual context.

### A.5.1 AGENT BEHAVIOR

We implemented a reactive agent that performs a random walk with wall-avoidance behavior. The agent has four possible actions: forward, left turn, right turn, and backward. The agent's behavior is characterized by:

- **Wall Detection**: The agent detects walls by attempting forward movement and checking if the position changes

- **Wall Avoidance**: When a wall is detected, the agent turns with $90\%$ probability (left or right with equal probability)

-

- **Forward Movement**: When no wall is detected, the agent always moves forward

- **Exploration**: With $10\%$ probability, the agent performs random turns even when no wall is present

The agent's heading is represented using both one-hot encoding and sinusoidal/cosinusoidal features, providing 6-dimensional heading information (4 one-hot + 2 sin/cos).

### A.5.2 SEQUENTIAL PREDICTION TASK

We formulate the task as $k$-step sequential prediction, where the model must predict future observations given:

- An initial observation $o_0$
- A sequence of $k$ future actions and heading features $f_{1:k} = \{a_t, h_t\}_{t=1}^{k}$

The model is trained to predict the corresponding sequence of future observations $o_{1:k}$ using mean squared error loss:

$$\mathcal{L} = \frac{1}{k} \sum_{t=1}^{k} \|o_t - \hat{o}_t\|_2^2 \tag{1}$$

where $\hat{o}_t$ is the model's prediction for observation at time $t$.

### A.5.3 MODEL ARCHITECTURE

We employ a GRU-based recurrent neural network with the following architecture:

- **Observation Encoder**: Two-layer MLP with hidden dimension $h = 128$ and ReLU activation, followed by a Tanh activation
- **Feature Encoder**: Two-layer MLP with the same architecture for processing action-heading features
- **GRU**: Two-layer GRU with hidden dimension $h = 128$
- **Prediction Head**: Two-layer MLP with ReLU activation for generating observation predictions

The model processes the initial observation $o_0$ through the observation encoder to initialize the GRU hidden state. The sequence of action-heading features $f_{1:k}$ is then processed through the feature encoder and fed to the GRU to generate predictions for each of the $k$ future time steps.

### A.5.4 TRAINING CONFIGURATION

The model is trained using the following hyperparameters:

- Training trajectories: 1000 trajectories of length $T = 100$
- Validation trajectories: 100 trajectories
- Batch size: 128
- Learning rate: $2 \times 10^{-3}$ (Adam optimizer)
- Gradient clipping: 1.0
- Training epochs: 10

We evaluate the model's learned representations using Principal Component Analysis (PCA) on the GRU hidden states, visualizing the 2D projection colored by the agent's $x$-position to assess spatial encoding capabilities.

### A.5.5 EVALUATION PROTOCOL

For evaluation, we generate 100 trajectories of length $T_{\text{eval}} = 50$ and extract hidden states from the trained model. We apply PCA to reduce the 128-dimensional hidden states to 2D for visualization, coloring points by the agent's $x$-position to reveal spatial structure in the learned representations.

The evaluation protocol allows us to assess whether the model has learned to encode spatial information in its hidden states, which would be evidenced by clustering or smooth transitions in the PCA visualization corresponding to the agent's position along the corridor.

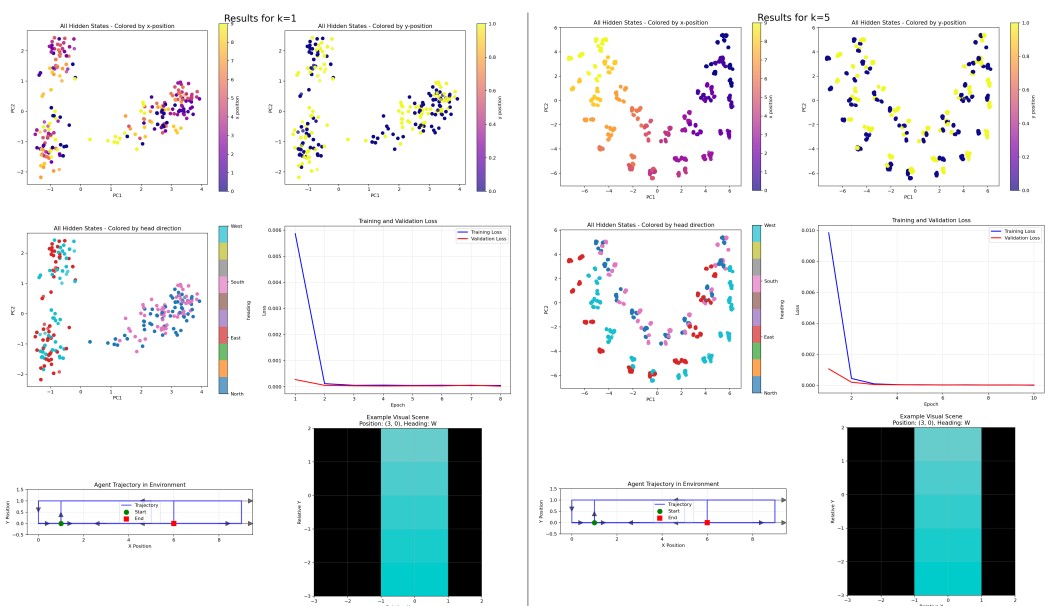

Figure S4: Detailed results for training an RNN on k-step prediction in a gridworld setting.

## A.6 PREDICTIVE CODING NETWORKS

We conducted simulations using predictive coding networks to investigate their performance on predictive learning tasks. All simulations were implemented in PyTorch based on this Github repository by Bogacz group.

### A.6.1 TASK DESIGN

The simulations employed a discrete state-action transition task where the network learned to predict the resulting state given a current state and action. States were represented as integers from $1$ to $S$, where $S = 20$ in all experiments. Actions were drawn from the range $[-A, A]$, where $A \in \{1, 10\}$ to examine performance under different action space complexities. For each valid state-action pair $(s, a)$, the target resulting state was computed as $s' = s + a$, subject to the constraint that $s' \in [1, S]$. This yielded a total of $N$ valid transitions, where $N$ varied with the action range $A$.

Inputs were encoded as concatenated one-hot vectors: the state was one-hot encoded into a vector of length $S$, and the action was one-hot encoded into a vector of length $2A + 1$ (mapping actions from $[-A, A]$ to indices $[0, 2A]$). The resulting input dimension was $S + 2A + 1$. The output was a probability distribution over the $S$ possible resulting states, with the task formulated as a multi-class classification problem using cross-entropy loss.

### A.6.2 NETWORK ARCHITECTURE

We trained feedforward neural networks with architectures ranging from 1 to 10 hidden layers. Each network consisted of:

- An input layer mapping from $S + 2A + 1$ dimensions to a hidden layer of size $H = 1048$
- Between 1 and 10 hidden layers, each of size $H = 1048$
- An output layer mapping from $H$ to $S$ dimensions

Each hidden layer was composed of a linear transformation, a predictive coding layer (`PCLayer`), and a ReLU activation function. The output layer consisted of a linear transformation without activation, producing logits for the $S$ output classes.

### A.6.3 PREDICTIVE CODING TRAINING

Networks were trained using the predictive coding framework, which minimizes an energy function through iterative inference. During training, for each batch:

1. **Inference phase**: The latent states $\mathbf{x}$ were updated for $T = 100$ iterations to minimize the energy function, which combines prediction errors (loss) and internal energy terms. Latent states were optimized using stochastic gradient descent (SGD) with learning rate $\eta_x = 0.01$.

2. **Parameter update phase**: After inference converged, network parameters $\boldsymbol{\theta}$ were updated using the Adam optimizer with learning rate $\eta_p = 0.001$. Parameters were updated only at the final inference step ($t = T$), following the standard predictive coding training protocol.

The energy function minimized during inference was:

$$E = \mathcal{L}(\mathbf{y}, \mathbf{y}^*) + \sum_{\ell} E_{\ell}(\mathbf{x}_{\ell}) \tag{2}$$

where $\mathcal{L}$ is the cross-entropy loss between predictions $\mathbf{y}$ and targets $\mathbf{y}^*$, and $E_{\ell}$ represents the energy associated with hidden layer $\ell$ containing latent states $\mathbf{x}_{\ell}$.

### A.6.4 TRAINING PROTOCOL

All networks were trained for 200 epochs using full-batch training (batch size equal to the number of valid transitions $N$). For each combination of network depth ($L \in \{1, 2, \ldots, 10\}$) and action range ($A \in \{1, 10\}$), we recorded:

- Accuracy (classification accuracy on the state prediction task)
- Loss (cross-entropy loss)
- Last hidden layer activations for principal component analysis (PCA)

### A.6.5 ANALYSIS METHODS

**Performance evaluation**: We computed classification accuracy and cross-entropy loss on the training set after each epoch. Since the task involved learning all valid transitions, training and test sets were identical (full dataset).

**Representation analysis**: To examine learned representations, we extracted hidden layer activations for all valid state-action pairs after training. We performed principal component analysis (PCA) on these activations. For visualization, we projected activations onto the first two principal components and colored data points according to: (1) output state, (2) input state, and (3) action value. When analyzing networks trained with $A > 1$, we filtered to transitions with $|a| \leq 1$ to generate comparable figures.

### A.6.6 CONVERGENCE MONITORING

In separate experiments, we monitored inference convergence during training to choose a verify the number of inference steps $T$ is sufficient. Convergence was assessed by tracking the relative change in loss over a sliding window of 5 consecutive inference steps. Inference was considered converged when the relative change fell below a threshold of $1\%$. This analysis was performed periodically during training (every 10 epochs) to characterize how inference dynamics evolve with learning, and to recommend optimal $T$ values based on the 90th percentile of observed convergence points.

## A.7 COMPUTATIONAL DETAILS

All simulations were implemented in Python 3 using PyTorch for neural network operations. The predictive coding framework was based on the implementation from the Bogacz Group *1_supervised_learning_pc.ipynb*.

