# OpenReview forum: "Multi-step Predictive Coding Leads To Simplicity Bias"
_ICLR.cc/2026/Conference — Submitted to ICLR 2026_

### Official Review · Reviewer_BphZ · 2025-10-27

**Soundness:** 2
**Presentation:** 3
**Contribution:** 2
**Rating:** 2
**Confidence:** 4

**Summary:**

This paper investigates how *multi-step prediction horizons* influence learned representations. Using a series of *toy environments*, the authors show that when models are trained to predict several steps into the future, the resulting internal representations tend to collapse onto low-dimensional manifolds that reflect the latent structure of the environment.

They analyze this phenomenon through:

- Experiments on a *deep linear model* trained on synthetic one-hot predictive tasks.
- A proposed connection between the model’s learned solutions and the Ordinary least squares estimator structure.
- Extensions to simple nonlinear settings, a piecewise-linear environment, and a toy MNIST variant where the model predicts digit transformations.

The main empirical observation is that *longer prediction horizons* induce smoother, lower-rank latent representations, while short-horizon prediction does not.

**Strengths:**

- **Clarity and organization:** The paper is well-written and easy to follow. The toy setups and visualizations (e.g., Figures 2–6) effectively illustrate the claimed phenomena.
- **Empirical insight:** It clearly shows that increasing prediction horizon can lead to emergent structure in latent representations, providing an interesting demonstration of implicit regularization in overparameterized systems.
- **Bridging empirical and analytical intuition:** The link drawn between OLS structure, gradient descent bias, and representation collapse is conceptually interesting, even if informal.

**Weaknesses:**

1. **Not actual predictive coding:**
Despite the framing, the models used are standard feedforward or GAN-like architectures trained with backpropagation on predictive tasks. There is no connection to biologically inspired predictive coding mechanisms or frameworks involving iterative inference or prediction error minimization. The title, abstract, and introduction should be rephrased to make this distinction clear and remove link to neuroscience.
2. **Toy nature of experiments:**
    - All the strong results occur in heavily idealized environments (one-hot states, small state spaces, or piecewise linear mappings).
    - The conclusions fail to generalize when within-class variability increases or when data complexity approaches real-world levels (as noted by the authors themselves for MNIST).
3. **Lack of theoretical depth:**
    - Theoretical analysis is *post hoc* and heuristic. The OLS and implicit bias discussions provide qualitative rather than rigorous proofs.
    - No new theorems or formal results are presented, only intuitive arguments.
4. **Limited novelty and impact:**
    - There is no compelling connection to either neuroscience (predictive coding) or impactful machine learning insights beyond toy-level illustration.

**Questions:**

1. How does your setup relate to predictive coding networks (e.g., Rao & Ballard 1999, Friston 2010, or modern hierarchical predictive coding frameworks)? Could your approach be reframed simply as multi-step supervised prediction?
2. Have you tested whether your results hold when using recurrent architectures trained on temporally structured data, rather than static feedforward mappings?
3. Could you provide *formal theoretical results* rather than qualitative explanations, e.g., a proof of when multi-step prediction induces rank compression or alignment with the latent manifold?
4. Given the limited generalization to realistic data (e.g., MNIST variability), what do you believe is the practical implication of your findings for representation learning or predictive modeling?

---

> ### Author Response · Authors · 2025-11-26
> **Part 1**
>
> We thank the reviewer for their thoughtful assessment and constructive
> suggestions. Below we address each point and describe the revisions we
> will implement.
>
> ## Relation to predictive coding.
>
> We agree that the term "predictive coding" may unintentionally evoke
> biologically motivated predictive coding networks (e.g.,\[6, 7\]). In
> this work we used it in the broader sense of *predictive learning
> tasks*, following prior work \[2-4\]. To avoid ambiguity, we will revise
> the paper to use terminology such as *predictive learning* or
> *predictive tasks*.
>
> Our setting can indeed be viewed as multi-step prediction, and we
> explicitly state this in the revised manuscript. However, this should
> not be conflated with supervised learning, as no explicit data labeling
> is required. Additionally, such predictive learning tasks can be
> implemented via biologically plausible mechanisms, such as delayed
> inputs and memory buffers.
>
> To address concerns regarding biological relevance, we additionally
> include new experiments using a Predictive Coding Network (PCN)
> architecture trained on the same task (Figure 7). The PCN exhibits the
> same qualitative dependence on prediction horizon, confirming that the
> effect is not tied to a particular optimization rule. The additional PCN
> experiment shows that, when framed as iterative prediction-error
> minimization, the same qualitative dependence on horizon emerges. This
> clarifies the relationship to PCN-style architectures without conflating
> the frameworks.
>
> ## Link to neuroscience.
>
> Despite the ambiguity noted above, our motivation is deeply rooted in
> neuroscience. First, we ground our work in a specific empirical
> observation from recent literature \[4\] and provide a theoretical
> explanation for it. Second, predictive learning frameworks---such as the
> Tolman-Eichenbaum machine \[1\] and the Successor Representation
> \[2\]---have been widely proposed to explain the origin of spatially
> tuned cells. Finally, we thank the reviewer for prompting us to make the
> neuroscientific implications of our work explicit. If the hippocampus
> indeed solves a long-horizon predictive task, an animal's exploration
> strategy must fundamentally shape the resulting representations. While
> this relationship was discussed early on regarding place cell
> directionality \[4\], it remains largely unexplored. Using the
> predictive learning framework, we aim to characterize how
> representations are shaped by behavior, cognitive variables, and sensory
> inputs. This understanding will allow us to explain why neural
> representations emerge with their specific properties and how they
> evolve under changing conditions. We plan to expand the discussion
> connecting the results from the paper to neuroscience.
>
> ## Toy nature of experiments and generality.
>
> We acknowledge that some experiments are intentionally minimalistic. Our
> focus in this submission was to identify *mechanisms* rather than
> empirical breadth. We use multiple settings to demonstrate that results
> generalize across modeling choices. We highlight the following results:
>
> -   **Recurrent models:** We already trained RNNs on temporally
>     structured gridworld trajectories (previously Appendix S4). These
>     results will be moved to the main text, demonstrating that the
>     effect is not restricted to static feedforward mappings. Note that
>     Levenstein et al. (2024) already performed extensive experiments in
>     that domain.
>
> -   **PCNs:** We trained PCNs on the predictive task and observed the
>     same qualitative dependence on prediction horizon (Figure 7).
>
> Together, these results demonstrate that the mechanism---interaction
> between horizon length and optimization-induced low-rank bias---is
> robust across architectures, observation types, and modeling choices.

---

> ### Author Response · Authors · 2025-11-27
> **Part 2**
>
> ## Theoretical depth.
>
> We appreciate the request for more formal theoretical development. While
> the paper does not present new rigorous analytical proofs, it connects
> several theoretical results to generate novel insights into the
> mechanism driving ordered representations. We acknowledge that the
> novelty of this connection was under-emphasized in the original
> manuscript.
>
> Linear networks classifying separable data (two classes) were shown to
> converge to the maximal margin solution \[8\]. In our work, we see that
> deep linear networks converge to suboptimal margins, seemingly in
> contradiction to the theory. Moreover, existing results claim maximum
> margin also for the multi-class case \[9\]. We combine multiple existing
> theoretical results to resolve this apparent contradiction. First, the
> solution for *multiclass* classification with homogeneous networks is
> the solution to a constrained optimization that maximizes margin while
> minimizing the norm of the weights. Second, the norm of the weights is a
> Frobenious norm across all layers, and another paper shows that this is
> equivalent to a Schatten quasi norm on the effective weights (the
> product of all layers) \[10\]. The implication of the Schatten norm is
> that deep networks are biased towards low-rank solutions, *at the
> expense* of maximizing the margin. The combination of these two papers
> explains the apparent contradiction with the first one.
>
> ## Practical implications and MNIST variability.
>
> We thank the reviewer for pointing out that our discussion of MNIST
> variability was unclear. We will revise the text to state that when
> within-class variability becomes too high, the latent state is no longer
> identifiable as a single dimension. In such cases, the predictive model
> may implicitly subdivide a digit class into multiple latent states
> (e.g., handwriting style, stroke curvature). Thus, lack of recovery of a
> single one-dimensional manifold in highly variable data reflects
> limitations of the latent generative structure rather than of the
> prediction-horizon principle itself. In real-world scenariosm
> variability of observations is generally constrained; furthermore, when
> a specific state is associated with very variable observations, it is
> reasonable to assume that they are not generated by a single state.
>
> We thank the reviewer again for the constructive and insightful
> comments. We believe that the revisions---terminology changes, added PCN
> and RNN experiments, clearer analytic discussion, and clarified MNIST
> interpretation---substantially strengthen the manuscript.
>
> *Note: Due to the short turnaround time, we have uploaded a preliminary
> revision containing the major changes. We are committed to fully
> polishing the final camera-ready version with all suggested changes.*
>
>
> [1] Whittington, J. C., et al. (2020).
>
> [2] Stachenfeld, K. L., et al.  (2017).
>
> [3] Recanatesi, S., et al. (2021).
>
> [4] Levenstein, D., et al. (2024).
>
> [5] Wood, E. R., et al. (2000).
>
> [6] Rao, R. P., & Ballard, D. H. (1999)
>
> [7] Friston, K. (2010)
>
> [8] Soudry, D., et al. (2018)
>
> [9] Lyu, K., & Li, J. (2019)
>
> [10] Dai, Z., et al. (2021)

---

### Official Review · Reviewer_gkNb · 2025-10-28

**Soundness:** 2
**Presentation:** 2
**Contribution:** 3
**Rating:** 4
**Confidence:** 3

**Summary:**

This paper explores the conditions under which neural networks learn simple representations of environments, capturing their structure.
The paper identifies a factor that contributes to this, namely the prediction horizon. In the paper this is formalized as the magnitude of the largest permitted action. The authors show that, increasing the magnitude of the largest permitted action helps the neural network learn simple representations of the training environment that capture its structure. This is shown in simple toy settings and in a more challenging task with MNIST images.

**Strengths:**

* The results for the GAN model are cool and creative. These also seem robust in the sense that several regularization methods were tried.
* The paper contains quite a few results (although some seem to be buried in the appendix?)

**Weaknesses:**

* It's not entirely clear whether action magnitude really captures multi-step prediction. Increasing the permissable action magnitude also offers more actions that can help the model disentangle the geometry of the environment. Is this really related to multi-step prediction?
* The line and grid environments are quite toyish, and other works have shown that other representation learning methods can disentangle environment geometry without multi-step prediction (see [1], [2] and [3]). These additional works should be discussed in the related works section.
* Some important details are not included in the main paper and discussed in the appendix. This is a bit strange since the paper is pretty short and comfortably below the page limit. In particular, the exposition of the line environment is very dense and difficult to understand. The same goes for the GAN experiment.

In my opinion, this is a good paper, but the exposition is a bit lacking. The experiments with the RNNs in the appendix can also be discussed more, as they offer an alternative setup to multi-step prediction than the one studied in section 3. The related works section also lacks references to past representation learning approaches that are able to learn good environment representations without multi-step prediction (see [1], [2] and [3]). Lastly, it has recently been shown that LLMs can learn representations of graphs in-context. These models have only been trained to perform next-token prediction, but nevertheless adapt to represent graph environments in a manner that captures their underlying structure (see [4] and [5]).
If the authors extend the related works section and make the exposition clearer I would be happy to increase my score to 6.

References:

[1] Watter et al. "Embed to control: A locally linear latent dynamics model for control from raw images." NeurIPS 2015

[2] Saanum et al. "Simplifying latent dynamics with softly state-invariant world models." NeurIPS 2024

[3] Kipf et al. "Contrastive learning of structured world models." ICLR 2020

[4] Park et al. "Iclr: In-context learning of representations." ICLR 2025

[5] Demircan et al. "Sparse Autoencoders Reveal Temporal Difference Learning in Large Language Models" ICLR 2025

**Questions:**

* Is there a reason for using a GAN in this experiment? Do similar representations emerge if you are using an Auto-encoder with a bottleneck to decode the target image label?
* When training the GAN with multi-step and single step actions, do you see differences in training dynamics and performance on the task it was trained to do? From what I can tell, only the representations were evaluated.

---

> ### Author Response · Authors · 2025-11-26
>
> We thank the reviewer for their constructive and detailed feedback.
> Below we address each concern and summarize the corresponding revisions
> to the manuscript.
>
> ## Connection between action magnitude and multi-step prediction.
>
> We appreciate the reviewer raising this conceptual point. In both
> discrete and continuous settings, actions can be provided in the
> original action space or as high-dimensional embeddings ("observations"
> of actions). A key result of this paper is that networks consistently
> recover the original action and state space from these high-dimensional
> embeddings. For these networks, prediction entails linearly combining
> the extracted state and action and then projecting the resulting state
> to the observation space. This holds for both temporal settings, where
> embedded actions are supplied consecutively ("multi-step"), and
> feedforward settings. We will explicitly clarify this connection in the
> revised manuscript.
>
> ## Related work.
>
> We thank the reviewer for highlighting several important prior works.
> The related work section has been expanded accordingly. Specifically, we
> note that while closely related to \[1-3\], our work provides a
> mechanistic explanation for why and when multi-step predictive coding
> alone---without additional regularization---reshapes the data geometry
> and consistently drives networks toward representations that recover the
> underlying latent state.
>
> Regarding \[4\], while the authors show that LLMs trained in single-step
> prediction recover the latent geometry of a lattice during in-context
> learning, they notably claim that representations linearly mirror the
> latent geometry only for a small $4\times4$ lattice. This is consistent
> with the scaling law observed in our work. For larger lattices, they
> demonstrate a transition point where the model learns the graph
> structure. However, this only requires learning local connections at
> each node rather than the global structure. It remains an open question
> whether linear global structure emerges for larger lattices and whether
> training with multi-step token prediction would facilitate this.
> Furthermore, the mapping between in-context learning \[4-5\] and
> prediction horizon is non-trivial. While LLMs are trained on
> single-token prediction, in-context learning involves behavioral changes
> derived from an accumulated context pool. In a sense, these models
> implicitly learn to plan multiple steps ahead based on an expanding
> context window.
>
> ## Clarity and placement of experimental details.
>
> We agree that several components of the original exposition belong in
> the main text. In the revised version, we will:
>
> -   Expand the explanation of the abstract line environment and rewrite
>     it for clarity.
>
> -   Move and expand the RNN experiments section, as the experiments
>     constitute an important demonstration of robustness.
>
> -   Clarify and expand the description of the GAN-based MNIST
>     experiment.
>
> ## Choice of GAN architecture.
>
> The use of a GAN is motivated both conceptually and practically. Because
> the input and output differ, autoencoding is not appropriate: the same
> input can map to multiple different targets. A GAN is meant to
> *generate* a sample from the distribution associated with the target
> class, which aligns with the goal of analyzing latent representations,
> rather than reconstructing specific images.
>
> ## Training dynamics and task performance.
>
> We thank the reviewer for pointing out the importance of evaluating
> predictive performance. We will include quantitative training curves for
> both single-step and multi-step GAN conditions. As expected, both
> conditions achieve similar prediction performance; the primary
> difference lies in the structure of the learned latent representations.
>
> We thank the reviewer once again for the thoughtful and constructive
> comments. The revised manuscript will include expanded related work,
> clarified exposition, additional experiments, and a clearer explanation
> of the conceptual connection between action magnitude and multi-step
> prediction.
>
> *Note: Due to the short turnaround time, we have uploaded a preliminary
> revision containing the major changes. We are committed to fully
> polishing the final camera-ready version with all suggested changes.*

---

### Official Review · Reviewer_kA7P · 2025-10-29

**Soundness:** 3
**Presentation:** 2
**Contribution:** 2
**Rating:** 4
**Confidence:** 1

**Summary:**

This paper looks at how multi-step predictive coding and using deeper neural networks can help models figure out the low-dimensional structure behind the data they’re observing. The authors start with a simple, theory-friendly setting and then move on to more complex and nonlinear tasks, such as experiments with MNIST digits. They show that when networks are deep enough and asked to predict farther into the future, their internal representations tend to organize themselves in ways that mirror the true structure of the environment. As a disclaimer, I'm definitely not an expert on the topic, so I will place low confidence.

**Strengths:**

The paper does a good job connecting theory and practical experiments. Starting simple, the authors gradually ramp up to more realistic datasets. Furthermore, the explanations around why deep networks and longer prediction horizons matter are clear and give new insights that go beyond what’s in earlier work, at least according to the author's claim.

The experiments extend to nonlinear, more natural examples, showing the main ideas hold up in several contexts. The theoretical claims of the authors seem well supported, and correct.

**Weaknesses:**

It seems to me that most of the results are for simple, linear cases. While results on more complex tasks are shown, it’s not totally clear how robust the main claims are in messier, real-world settings. I would be interested in seeing experiments on slightly more complex tasks.

**Questions:**

Overall, the paper is well-written and the results are intriguing, but maybe there’s still room for more evidence about what happens in tougher settings or with newer architectures. How come the authors did not test on this setup?

---

> ### Author Response · Authors · 2025-11-26
>
> We thank the reviewer for the thoughtful and constructive feedback, as
> well as for highlighting the clarity of our exposition, the connection
> between theoretical insights and empirical results, and the novelty of
> our core observations. We also appreciate the reviewer's disclaimer
> regarding confidence and hope that the clarifications and new results
> detailed below will strengthen the assessment.
>
> ## Robustness beyond simple linear settings.
>
> We agree that while the clearest mechanistic intuition arises in linear
> and controlled settings---a design choice intended to isolate the
> underlying principle tractably---assessing the robustness of the results
> is crucial. In the revised manuscript, we have expanded this aspect of
> the work:
>
> -   **Predictive Coding Networks (PCNs).** We now include results using
>     modern predictive coding networks trained on the same multi-step
>     predictive tasks. PCNs exhibit the *same qualitative dependence* on
>     the prediction horizon, demonstrating that the mechanism is not tied
>     exclusively to backpropagation (Figure 7).
>
> -   **Recurrent sequential environments.** We will move the RNN
>     gridworld prediction experiment (previously in the appendix) to the
>     main text. These results demonstrate that multi-step prediction
>     leads to structured representations even under temporally correlated
>     sequential input.
>
> Together, these additions demonstrate that the emergence of latent
> structure under multi-step prediction is *not* specific to toy settings,
> linear networks, or feedforward architectures. Rather, it persists
> across PCNs, CNNs, GANs, and RNNs.
>
> ## Realistic or "messier" environments.
>
> We agree that extending the study to very large-scale real-world
> datasets is an important future direction. However, our focus in this
> submission was to identify *mechanisms* rather than maximize empirical
> breadth. The scaling behavior we uncover provides a principled
> perspective for understanding *when* latent structures can be recovered.
> The purpose of testing different settings was to demonstrate the
> robustness of the main observation to modeling choices rather than to
> showcase deployment in unconstrained real-world scenarios.
>
> We thank the reviewer again for the positive evaluation of the paper's
> clarity and insights. We hope that the expanded experiments and improved
> exposition address the concerns and highlight the robustness of our
> findings.
>
> *Note: Due to the short turnaround time, we have uploaded a preliminary
> revision containing the major changes. We are committed to fully
> polishing the final camera-ready version with all suggested changes.*

---

### Author Response · Authors · 2025-12-02
**Message to the new Area Chair**

Thank you for taking over the evaluation of our submission. We would like to briefly clarify the intent of the paper and provide context for the reviews, as the reviewer–author discussion unfortunately ended before the reviewers could respond to several critical misunderstandings.

### **Scope and intent of the paper**
This paper presents theoretical and mechanistic work aimed at understanding why increasing the prediction horizon in a predictive task leads neural networks to recover the latent geometry of an environment—a phenomenon observed in recent neuroscience and machine learning studies. Our goal is not to propose a new method or improve benchmark performance. Rather, we seek to identify the underlying mechanism in settings that are analytically and conceptually tractable.

Because of this focus, we deliberately use controlled “toy” environments that allow us to expose the mechanism clearly, and then demonstrate its robustness across a range of modeling choices (linear networks, nonlinear networks, CNNs, GANs, RNNs). Following the review, we also added results with a Predictive Coding Network, further demonstrating that the effect can be demonstrated in a biologically plausible setting.

### **Our core research question**
Does increasing the prediction horizon have a general and consistent effect on representations, and if so, why?
The paper shows that the answer is yes: longer horizons produce a complex higher-dimensional predictive task, but also a correlation structure that is dominated by low-dimensional directions reflecting the latent state geometry. Deep networks with low-rank bias consistently recover this structure. This mechanistic explanation is relevant for predictive learning in ML and for understanding hippocampal representations in neuroscience.

### **Regarding the reviews**
Because rebuttal discussion was blocked early, we could not get reviewer feedback to our clarifications. We summarize for completeness:

**Reviewer 1** explicitly indicated a confidence of 1, signaling that their review should not strongly influence the final decision.

**Reviewer 2** stated that they would raise their score after expanding the related-works section, which we have done. We also addressed all additional questions regarding architecture choice, alternative models, and the meaning of action magnitude.

**Reviewer 3** provided the highest-confidence and most thorough review. The reviewer correctly noted that we do not employ “predictive coding’’ in the sense of a free-energy–based learning algorithm. We did not claim to do so; our use of the term referred to solving a predictive task. We have updated the terminology throughout the paper and added a section demonstrating that a Predictive Coding Network (PCN) also solves our predictive task and exhibits the same qualitative behavior. This resolves the terminological confusion and shows that our results are relevant to the predictive-coding literature. We believe we have addressed most of the reviewer’s concerns, including relevance to neuroscience, practical implications, and generality of results.

The remaining point of concern would likely have been the level of theoretical depth. We emphasize that our contribution lies in connecting several existing theoretical results and combining them to reveal a previously unrecognized implicit bias of deep linear networks solving multi-class classification tasks with separable data. While we agree the paper does not contain full rigorous proofs, it provides substantial analytical insight by synthesizing prior theoretical work with new numerical evidence.

---

### Meta-Review · Area_Chair_TriL · 2026-01-07

**Summary:**

Paper studies why longer prediction horizons + sufficient depth induce simple/low-rank representations that recover latent environment geometry. It combines controlled “toy” predictive tasks with extensions (nonlinear nets, GAN/MNIST variant, RNN gridworld) and provides a mechanistic story via OLS structure + implicit low-rank bias in deep linear networks. Reviews are mixed: two marginal-below (kA7P=4 low conf; gkNb=4), one reject (BphZ=2). Core disagreements are about framing (“predictive coding”), generality beyond toys, and theoretical rigor/novelty.

**Reviewer Concerns:**

What rebuttal addressed:

Terminology/framing: authors agree “predictive coding” was ambiguous; commit to reframe as predictive learning / multi-step prediction, and add Predictive Coding Network (PCN) experiments showing the same horizon effect (helps answer BphZ’s main critique).

Generality: move RNN gridworld results to main; emphasize robustness across CNNs/GANs/RNNs/PCNs.

Related work: expand to include representation-learning methods that recover geometry without multi-step prediction (Watter/Kipf/etc.) and situate contribution as mechanistic explanation rather than “first to do X”.

Clarity/space usage: promise to pull key appendix details (line env, GAN, RNN) into main and improve exposition; add training curves for single vs multi-step GAN.

Theory positioning: clarify they synthesize existing theory (margin + Schatten quasi-norm / low-rank bias) rather than claiming new formal theorems.

Remaining concerns:

Framing & scope: even with terminology fixes, some reviewers may still view the contribution as toy-mechanism demonstration with limited practical ML impact; neuroscience relevance may read speculative without stronger empirical linkage.

Action magnitude vs “multi-step”: authors explain, but the conceptual equivalence may remain non-obvious to readers; could still feel like a confound (more powerful actions ≠ longer horizon).

Theory depth: explanation remains largely heuristic/synthesis; no new tight conditions/proofs for when rank compression/latent recovery is guaranteed.

Empirical breadth: MNIST results appear fragile to within-class variability; “messier” real-world settings still not tested.

**Reviewer Scores:**

Score-change guess (post-rebuttal):

kA7P (conf=1): 4 → 4–5 (added PCN/RNN helps but they already signaled low confidence)

gkNb: 4 → 5 (they explicitly said related work + clarity + more on RNNs could justify a raise; rebuttal commits to all)

BphZ: 2 → 3 (major “not predictive coding” issue is directly addressed via terminology + PCN experiment; but concerns on toy/generalization/theory likely remain)

---

### Decision · Program_Chairs · 2026-01-26

Reject